# Improving Relation Extraction by Pre-trained Language Representations

**Christoph Alt**[*]                                              CHRISTOPH.ALT@DFKI.DE

**Marc Hübner**[*]                                              MARC.HUEBNER@DFKI.DE

**Leonhard Hennig**                                          LEONHARD.HENNIG@DFKI.DE

*German Research Center for Artificial Intelligence (DFKI)*

*Speech and Language Technology Lab, Alt-Moabit 91c, 10559 Berlin, Germany*

## Abstract

Current state-of-the-art relation extraction methods typically rely on a set of lexical, syntactic, and semantic features, explicitly computed in a pre-processing step. Training feature extraction models requires additional annotated language resources, which severely restricts the applicability and portability of relation extraction to novel languages. Similarly, pre-processing introduces an additional source of error. To address these limitations, we introduce TRE, a Transformer for Relation Extraction, extending the OpenAI Generative Pre-trained Transformer [Radford et al., 2018]. Unlike previous relation extraction models, TRE uses pre-trained deep language representations instead of explicit linguistic features to inform the relation classification and combines it with the self-attentive Transformer architecture to effectively model long-range dependencies between entity mentions. TRE allows us to learn implicit linguistic features solely from plain text corpora by unsupervised pre-training, before fine-tuning the learned language representations on the relation extraction task. TRE obtains a new state-of-the-art result on the TACRED and SemEval 2010 Task 8 datasets, achieving a test F1 of 67.4 and 87.1, respectively. Furthermore, we observe a significant increase in sample efficiency. With only 20% of the training examples, TRE matches the performance of our baselines and our model trained from scratch on 100% of the TACRED dataset. We open-source our experiments and code[1].

## 1. Introduction

Relation extraction aims to identify the relationship between two nominals, typically in the context of a sentence, making it essential to natural language understanding. Consequently, it is a key component of many natural language processing applications, such as information extraction [Fader et al., 2011], knowledge base population [Ji and Grishman, 2011], and question answering [Yu et al., 2017]. Table 1 lists exemplary relation mentions.

State-of-the-art relation extraction models, traditional feature-based and current neural methods, typically rely on explicit linguistic features: prefix and morphological features [Mintz et al., 2009], syntactic features such as part-of-speech tags [Zeng et al., 2014], and semantic features like named entity tags and WordNet hypernyms [Xu et al., 2016]. Most recently, Zhang et al. [2018] combined dependency parse features with graph convolutional neural networks to considerably increase the performance of relation extraction systems.

---

*. equal contribution

1. https://github.com/DFKI-NLP/TRE

| Sentence | Subject | Object | Relation |
|---|---|---|---|
| Mr. Scheider played the police chief of a resort town menaced by a shark. | Scheider | police chief | per:title |
| The measure included Aerolineas's domestic subsidiary, Austral. | Aerolineas | Austral | org:subsidiaries |
| Yolanda King died last May of an apparent heart attack. | Yolanda King | heart attack | per:cause_of_death |
| The key was in a chest. | key | chest | Content-Container |
| The car left the plant. | car | plant | Entity-Origin |
| Branches overhang the roof of this house. | roof | house | Component-Whole |

Table 1: Relation extraction examples, taken from TACRED (1-3) and SemEval 2010 Task 8 (4-6). TACRED relation types mostly focus on named entities, whereas SemEval contains semantic relations between concepts.

However, relying on explicit linguistic features severely restricts the applicability and portability of relation extraction to novel languages. Explicitly computing such features requires large amounts of annotated, language-specific resources for training; many unavailable in non-English languages. Moreover, each feature extraction step introduces an additional source of error, possibly cascading over multiple steps. Deep language representations, on the other hand, have shown to be a very effective form of unsupervised pre-training, yielding contextualized features that capture linguistic properties and benefit downstream natural language understanding tasks, such as semantic role labeling, coreference resolution, and sentiment analysis [Peters et al., 2018]. Similarly, fine-tuning pre-trained language representations on a target task has shown to yield state-of-the-art performance on a variety of tasks, such as semantic textual similarity, textual entailment, and question answering [Radford et al., 2018].

In addition, classifying complex relations requires a considerable amount of annotated training examples, which are time-consuming and costly to acquire. Howard and Ruder [2018] showed language model fine-tuning to be a sample efficient method that requires fewer labeled examples.

Besides recurrent (RNN) and convolutional neural networks (CNN), the Transformer [Vaswani et al., 2017] is becoming a popular approach to learn deep language representations. Its self-attentive structure allows it to capture long-range dependencies efficiently; demonstrated by the recent success in machine translation [Vaswani et al., 2017], text generation [Liu et al., 2018], and question answering [Radford et al., 2018].

In this paper, we propose TRE: a **T**ransformer based **R**elation **E**xtraction model. Unlike previous methods, TRE uses deep language representations instead of explicit linguistic features to inform the relation classifier. Since language representations are learned by unsupervised language modeling, pre-training TRE only requires a plain text corpus instead of annotated language-specific resources. Fine-tuning TRE, and its representations, directly

to the task minimizes explicit feature extraction, reducing the risk of error accumulation. Furthermore, an increased sample efficiency reduces the need for distant supervision methods [Mintz et al., 2009, Riedel et al., 2010], allowing for simpler model architectures without task-specific modifications.

The contributions of our paper are as follows:

- We describe TRE, a Transformer based relation extraction model that, unlike previous methods, relies on deep language representations instead of explicit linguistic features.

- We are the first to demonstrate the importance of pre-trained language representations in relation extraction, by outperforming state-of-the-art methods on two supervised datasets, TACRED and SemEval 2010 Task 8.

- We report detailed ablations, demonstrating that pre-trained language representations prevent overfitting and achieve better generalization in the presence of complex entity mentions. Similarly, we demonstrate a considerable increase in sample efficiency over baseline methods.

- We make our trained models, experiments, and source code available to facilitate wider adoption and further research.

## 2. TRE

This section introduces TRE and its implementation. First, we cover the model architecture (Section 2.1) and input representation (Section 2.2), followed by the introduction of unsupervised pre-training of deep language model representations (Section 2.3). Finally, we present supervised fine-tuning on the relation extraction task (Section 2.4).

### 2.1 Model Architecture

TRE is a multi-layer Transformer-Decoder [Liu et al., 2018], a decoder-only variant of the original Transformer [Vaswani et al., 2017]. As shown in Figure 1, the model repeatedly encodes the given input representations over multiple layers (i.e., Transformer blocks), consisting of masked multi-headed self-attention followed by a position-wise feedforward operation:

$$h_0 = TW_e + W_p$$
$$h_l = transformer\_block(h_{l-1}) \ \forall \ l \ \in \ [1, L] \tag{1}$$

Where $T = (t_1, \ldots, t_k)$ is a sequence of token indices in a sentence[2]. $W_e$ is the token embedding matrix, $W_p$ is the positional embedding matrix, $L$ is the number of Transformer blocks, and $h_l$ is the state at layer $l$. Since the Transformer has no implicit notion of token positions, the first layer adds a learned positional embedding $e_p \in \mathbb{R}^d$ to each token embedding $e_t^p \in \mathbb{R}^d$ at position $p$ in the input sequence. The self-attentive architecture allows an output state $h_l^p$ of a block to be informed by all input states $h_{l-1}$, which is key to efficiently model long-range dependencies. For language modeling, however, self-attention

---

2. In this work a "sentence" denotes an arbitrary span of contiguous text, rather than an actual linguistic sentence. For pre-training the input consists of multiple linguistic sentences, whereas relation extraction is applied to a single one. A "sequence" refers to the input token sequence.

must be constrained (masked) not to attend to positions ahead of the current token. For a more exhaustive description of the architecture, we refer readers to Vaswani et al. [2017] and the excellent guide "The Annotated Transformer"[3].

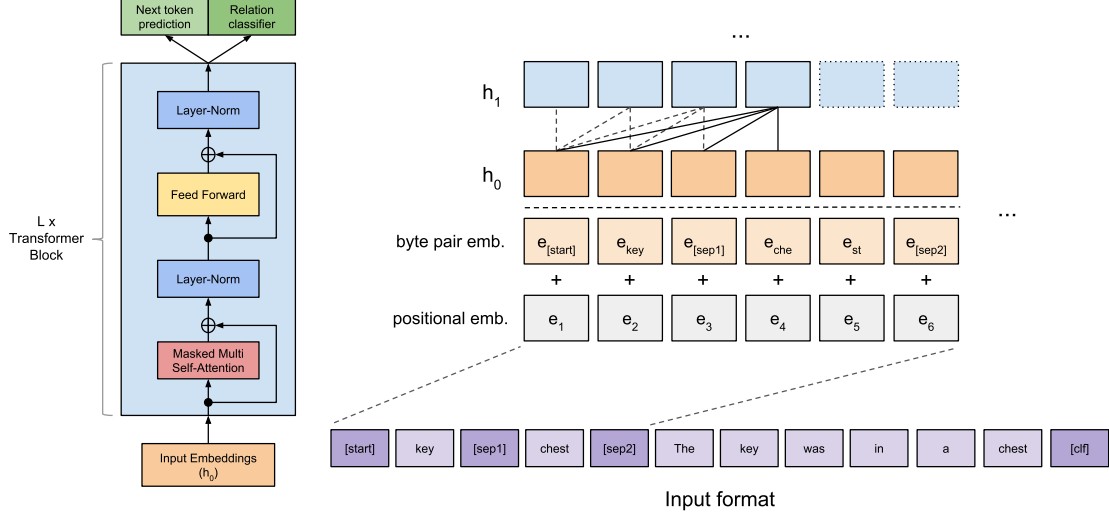

Figure 1: **(left)** Transformer-Block architecture and training objectives. A Transformer-Block is applied at each of the $L$ layers to produce states $h_1$ to $h_L$. **(right)** Relation extraction requires a structured input for fine-tuning, with special delimiters to assign different meanings to parts of the input.

## 2.2 Input Representation

Our input representation (see also Figure 1) encodes each sentence as a sequence of tokens. To make use of sub-word information, we tokenize the input text using byte pair encoding (BPE) [Sennrich et al., 2016]. The BPE algorithm creates a vocabulary of sub-word tokens, starting with single characters. Then, the algorithm iteratively merges the most frequently co-occurring tokens into a new token until a predefined vocabulary size is reached. For each token, we obtain its input representation by summing over the corresponding token embedding and positional embedding.

While the model is pre-trained on plain text sentences, the relation extraction task requires a structured input, namely a sentence and relation arguments. To avoid task-specific changes to the architecture, we adopt a traversal-style approach similar to Radford et al. [2018]. The structured, task-specific input is converted to an ordered sequence to be directly fed into the model without architectural changes. Figure 1 provides a visual illustration of the input format. It starts with the tokens of both relation arguments $a^1$ and $a^2$, separated by delimiters, followed by the token sequence of the sentence containing the relation mention, and ends with a special classification token. The classification token signals the model to generate a sentence representation for relation classification. Since our

---

3. http://nlp.seas.harvard.edu/2018/04/03/attention.html

model processes the input left-to-right, we add the relation arguments to the beginning, to bias the attention mechanism towards their token representation while processing the sentence's token sequence.

### 2.3 Unsupervised Pre-training of Language Representations

Relation extraction models benefit from efficient representations of long-term dependencies [Zhang et al., 2018] and hierarchical relation types [Han et al., 2018]. Generative pre-training via a language model objective can be seen as an ideal task to learn deep language representations that capture important lexical, syntactic, and semantic features without supervision [Linzen et al., 2016, Radford et al., 2018, Howard and Ruder, 2018], before fine-tuning to the supervised task – in our case relation extraction.

Given a corpus $\mathcal{C} = \{c_1, \ldots, c_n\}$ of tokens $c_i$, the language modeling objective maximizes the likelihood

$$L_1(\mathcal{C}) = \sum_i \log P(c_i | c_{i-1}, \ldots, c_{i-k}; \theta) \tag{2}$$

where k is the context window considered for predicting the next token $c_i$ via the conditional probability $P$. TRE models the conditional probability by an output distribution over target tokens:

$$P(c) = softmax(h_L W_e^T), \tag{3}$$

where $h_L$ is the sequence of states after the final layer $L$, $W_e$ is the embedding matrix, and $\theta$ are the model parameters that are optimized by stochastic gradient descent.

### 2.4 Supervised Fine-tuning on Relation Extraction

After pre-training with the objective in Eq. 2, the language model is fine-tuned on the relation extraction task. We assume a labeled dataset $\mathcal{D} = ([x_i^1, \ldots, x_i^m], a_i^1, a_i^2, r_i)$, where each instance consists of an input sequence of tokens $x^1, \ldots, x^m$, the positions of both relation arguments $a^1$ and $a^2$ in the sequence of tokens, and a corresponding relation label $r$. The input sequence is fed to the pre-trained model to obtain the final state representation $h_L$. To compute the output distribution $P(r)$ over relation labels, a linear layer followed by a softmax is applied to the last state $h_L^m$, which represents a summary of the input sequence:

$$P(r | x^1, \ldots, x^m) = softmax(h_L^m W_r) \tag{4}$$

During fine-tuning we optimize the following objective:

$$L_2(\mathcal{D}) = \sum_{i=1}^{|\mathcal{D}|} \log P(r_i | x_i^1, \ldots, x_i^m) \tag{5}$$

According to Radford et al. [2018], introducing language modeling as an auxiliary objective during fine-tuning improves generalization and leads to faster convergence. Therefore, we adopt a similar objective:

$$L(\mathcal{D}) = \lambda * L_1(\mathcal{D}) + L_2(\mathcal{D}), \tag{6}$$

where $\lambda$ is the language model weight, a scalar, weighting the contribution of the language model objective during fine-tuning.

## 3. Experiment Setup

We run experiments on two supervised relation extraction datasets: The recently published TACRED dataset [Zhang et al., 2017] and the SemEval dataset [Hendrickx et al., 2010]. We evaluate the PCNN implementation of Zeng et al. [2015][4] on the two datasets and use it as a state-of-the-art baseline in our analysis section. In the following we describe our experimental setup.

### 3.1 Datasets

Table 2 shows key statistics of the TACRED and SemEval datasets. While TACRED is approximately 10x the size of SemEval 2010 Task 8, it contains a much higher fraction of negative training examples, which makes classification more challenging.

| Dataset | relation types | examples | negative examples |
|---|---|---|---|
| TACRED | 42 | 106,264 | 79.5% |
| SemEval 2010 Task 8 | 19 | 10,717 | 17.4% |

Table 2: Comparison of the datasets used for evaluation

**TACRED**: The *TAC Relation Extraction Dataset* [Zhang et al., 2017] contains 106k sentences with entity mention pairs collected from the TAC KBP evaluations[5] 2009–2014, with the years 2009 to 2012 used for training, 2013 for evaluation, and 2014 for testing. Sentences are annotated with person- and organization-oriented relation types, e.g. *per:title*, *org:founded*, and *no_relation* for negative examples. In contrast to the SemEval dataset the entity mentions are typed, with subjects classified into person and organization, and objects categorized into 16 fine-grained classes (e.g., date, location, title). As per convention, we report our results as micro-averaged F1 scores. Following the evaluation strategy of Zhang et al. [2017], we select our best model based on the median validation F1 score over 5 independent runs and report its performance on the test set.

**SemEval 2010 Task 8**: The SemEval 2010 Task 8 dataset [Hendrickx et al., 2010] is a standard benchmark for binary relation classification, and contains 8,000 sentences for training, and 2,717 for testing. Sentences are annotated with a pair of untyped nominals and one of 9 directed semantic relation types, such as *Cause-Effect*, *Entity-Origin*, as well as the undirected *Other* type to indicate no relation, resulting in 19 distinct types in total. We follow the official convention and report macro-averaged F1 scores with directionality taken into account by averaging over 5 independent runs. The dataset is publicly available[6] and we use the official scorer to compute our results.

### 3.2 Pre-training

Since pre-training is computationally expensive, and our main goal is to show its effectiveness by fine-tuning on the relation extraction task, we reuse the language model[7] published

---

4. https://github.com/thunlp/OpenNRE

5. https://tac.nist.gov/2017/KBP/index.html

6. https://drive.google.com/file/d/0B_jQiLugGTAkMDQ5ZjZiMTUtMzQ1Yy00YWNmLWJlZDYtOWY1ZDMwY2U4YjFk

7. https://github.com/openai/finetune-transformer-lm

by Radford et al. [2018] for our experiments. The model was trained on the BooksCorpus [Zhu et al., 2015], which contains around 7,000 unpublished books with a total of more than 800M words of different genres. It consists of $L = 12$ layers (blocks) with 12 attention heads and 768 dimensional states, and a feed-forward layer of 3072 dimensional states. We reuse the model's byte pair encoding vocabulary, containing 40,000 tokens, but extend it with task-specific ones (i.e., start, end, and delimiter tokens). Also, we use the learned positional embeddings with supported sequence lengths of up to 512 tokens.

### 3.3 Entity Masking

We employ four different entity masking strategies. Entity masking allows us to investigate the model performance while providing limited information about entities, in order to prevent overfitting and allow for better generalization to unseen entities. It also enables us to analyze the impact of entity type and role features on the model's performance. For the simplest masking strategy *UNK*, we replace all entity mentions with a special unknown token. For the *NE* strategy, we replace each entity mention with its named entity type. Similarly, *GR* substitutes a mention with its grammatical role (subject or object). *NE + GR* combines both strategies.

### 3.4 Hyperparameter Settings and Optimization

During our experiments we found the hyperparameters for fine-tuning, reported in [Radford et al., 2018], to be very effective. Therefore, unless specified otherwise, we used the Adam optimization scheme [Kingma and Ba, 2015] with $\beta_1 = 0.9$, $\beta_2 = 0.999$, a batch size of 8, and a linear learning rate decay schedule with warm-up over 0.2% of training updates. We apply residual, and classifier dropout with a rate of 0.1. Also, we experimented with token dropout, but did not find that it improved performance. Table 3 shows the best performing hyperparameter configuration for each dataset. On SemEval 2010 Task 8, we first split 800 examples of the training set for hyperparameter selection and retrained on the entire training set with the best parameter configuration.

|  | Epochs | Learning Rate | Warmup Learning Rate | $\lambda$ | Attn. Dropout |
|---|---|---|---|---|---|
| TACRED | 3 | 5.25e-5 | 2e-3 | 0.5 | 0.1 |
| SemEval | 3 | 6.25e-5 | 1e-3 | 0.7 | 0.15 |

Table 3: Best hyperparameter configuration for TACRED and SemEval

## 4. Results

This section presents our experimental results. We compare our TRE model to other works on the two benchmark datasets, demonstrating that it achieves state-of-the-art performance even without sophisticated linguistic features. We also provide results on model ablations and the effect of the proposed entity masking schemes.

### 4.1 TACRED

On the TACRED dataset, TRE outperforms state-of-the-art single-model systems and achieves an F1 score of 67.4 (Table 4). Compared to SemEval, we observe methods to perform better that are able to model complex syntactic and long-range dependencies, such as PA-LSTM [Zhang et al., 2017] and C-GCN [Zhang et al., 2018]. Outperforming these methods highlights our model's ability to implicitly capture patterns similar to complex syntactic features, and also capture long-range dependencies.

We would like to point out that the result was produced by the same "entity masking" strategy used in previous work [Zhang et al., 2017, 2018]. Similar to our *NE + GR* masking strategy, described in Section 3.3, we replace each entity mention by a special token; a combination of its named entity type and grammatical role. While we achieve state-of-the-art results by providing only named entity information, unmasked entity mentions decrease the score to 62.8, indicating overfitting and, consequently, difficulties to generalize to specific entity types. In Section 5.3, we analyze the effect of entity masking on task performance in more detail.

| System | P | R | F1 |
|---|---|---|---|
| LR† Zhang et al. [2017] | 72.0 | 47.8 | 57.5 |
| CNN† Zhang et al. [2017] | **72.1** | 50.3 | 59.2 |
| Tree-LSTM† Zhang et al. [2018] | 66.0 | 59.2 | 62.4 |
| PA-LSTM† Zhang et al. [2018] | 65.7 | 64.5 | 65.1 |
| C-GCN† Zhang et al. [2018] | 69.9 | 63.3 | 66.4 |
| TRE (ours) | 70.1 | **65.0** | **67.4** |

Table 4: TACRED single-model test set performance. We selected the hyperparameters using the validation set, and report the test score of the run with the median validation score among 5 randomly initialized runs. † marks results reported in the corresponding papers.

### 4.2 SemEval

On the SemEval 2010 Task 8 dataset, the TRE model outperforms the best previously reported models, establishing a new state-of-the-art score of 87.1 F1 (Table 5). The result indicates that pre-training via a language modeling objective allows the model to implicitly capture useful linguistic properties for relation extraction, outperforming methods that rely on explicit lexical features (SVM [Rink and Harabagiu, 2010], RNN [Zhang and Wang, 2015]). Similarly, our model outperforms approaches that rely on explicit syntactic features such as the shortest dependency path and learned distributed representations of part-of-speech and named entity tags (e.g., BCRNN [Cai et al., 2016], DRNN [Xu et al., 2016], CGCN [Zhang et al., 2018]).

Similar to Zhang et al. [2018], we observe a high correlation between entity mentions and relation labels. According to the authors, simplifying SemEval sentences in the training and validation set to just *"subject and object"*, where "subject" and "object" are the actual entities, already achieves an F1 score of 65.1. To better evaluate our model's ability to

| System | P | R | F1 |
|---|---|---|---|
| SVM[†] Rink and Harabagiu [2010] | – | – | 82.2 |
| PA-LSTM[†] Zhang et al. [2018] | – | – | 82.7 |
| C-GCN[†] Zhang et al. [2018] | – | – | 84.8 |
| DRNN[†] Xu et al. [2016] | – | – | 86.1 |
| BRCNN[†] Cai et al. [2016] | – | – | 86.3 |
| PCNN Zeng et al. [2015] | 86.7 | 86.7 | 86.6 |
| TRE (ours) | 88.0 | 86.2 | **87.1** ($\pm$0.16) |

Table 5: SemEval single-model test set performance. † marks results reported in the corresponding papers. We report the mean and standard deviation across 5 randomly initialized runs.

generalize beyond entity mentions, we substitute the entity mentions in the training set with a special unknown *(UNK)* token. The token simulates the presence of unseen entities and prevents overfitting to entity mentions that strongly correlate with specific relations. Our model achieves an F1 score of 79.1 (Table 6), an improvement of 2.6 points F1 score over

| System | P | R | F1 |
|---|---|---|---|
| PA-LSTM[†] Zhang et al. [2018] | – | – | 75.3 |
| C-GCN[†] Zhang et al. [2018] | – | – | 76.5 |
| TRE (ours) | 80.3 | 78.0 | **79.1** ($\pm$ 0.37) |

Table 6: SemEval single-model test set performance with all entity mentions masked by an unknown *(UNK)* token. † marks results reported in the corresponding papers. Due to the small test set size, we report the mean and standard deviation across 5 randomly initialized runs.

the previous state-of-the-art. The result suggests that pre-trained language representations improve our model's ability to generalize beyond the mention level when predicting the relation between two previously unseen entities.

## 5. Analysis & Ablation Studies

Although we demonstrated strong empirical results, we have not yet isolated the contributions of specific parts of TRE. In this section, we perform ablation experiments to understand the relative importance of each model component, followed by experiments to validate our claim that pre-trained language representations capture linguistic properties useful to relation extraction and also improve sample efficiency. We report our results on the predefined TACRED validation set and randomly select 800 examples of the SemEval training set as a validation set.

### 5.1 Effect of Pre-training

Pre-training affects two major parts of our model: language representations and byte pair embeddings. In Table 7, we first compare a model that was fine-tuned using pre-trained representations to one that used randomly initialized language representations. On both datasets we observe fine-tuning to considerably benefit from pre-trained language representations. For the SemEval dataset, the validation F1 score increases to 85.6 when using a pre-trained language model and no entity masking, compared to 75.6 without pre-training. We observe even more pronounced performance gains for the TACRED dataset, where using a pre-trained language model increases the validation F1 score by 20 to 63.3. With entity masking, performance gains are slightly lower, at +8 on the SemEval dataset and +9.4 (UNK) respectively +3.8 (NE+GR) on the TACRED dataset. The larger effect of pre-training when entity mentions are not masked suggests that pre-training has a regularizing effect, preventing overfitting to specific mentions. In addition, the contextualized features allow the model to better adapt to complex entities. Our observations are consistent with the results of Howard and Ruder [2018], who observed that language model pre-training considerably improves text classification performance on small and medium-sized datasets, similar to ours.

| | SemEval | | TACRED | | |
|---|---|---|---|---|---|
| | None | UNK | None | UNK | NE + GR |
| Best model | **85.6** | 76.9 | 63.3 | 51.0 | **68.0** |
| – w/o pre-trained LM | **75.6** | 68.2 | 43.3 | 41.6 | **64.2** |
| – w/o pre-trained LM and BPE | 55.3 | **60.9** | 38.5 | 38.4 | **60.8** |

Table 7: Ablation with and without masked entities for SemEval **(left)** and TACRED validation set **(right)**. We report F1 scores over 5 independent runs.

In addition, we train a model from scratch without pre-trained byte pair embeddings. We keep the vocabulary of sub-word tokens fixed and randomly initialize the embeddings. Again, we observe both datasets to benefit from pre-trained byte-pair embeddings. Because of its small size, SemEval benefits more from pre-trained embeddings, as these can not be learned reliably from the small corpus. This increases the risk of overfitting to entity mentions, which can be seen in the lower performance compared to UNK masking, where entity mentions are not available. For the TACRED dataset, model performance drops by approximately $3 - 5\%$ with and without entity masking when not using pre-trained byte pair embeddings.

### 5.2 Which Information is captured by Language Representations?

Undoubtedly, entity type information is crucial to relation extraction. This is confirmed by the superior performance on TACRED (Table 7) when entity and grammatical role information is provided (NE+GR). The model achieves a validation F1 score of 68.0, compared to 63.3 without entity masking. Without pre-trained language representations, the model with NE+GR masking still manages to achieve a F1 score of 64.2. This suggests that pre-trained

language representations capture features that are as informative as providing entity type and grammatical role information. This is also suggested by the work of Peters et al. [2018], who show that a language model captures syntactic and semantic information useful for a variety of natural language processing tasks such as part-of-speech tagging and word sense disambiguation.

### 5.3 Effect of Entity Masking

Entity masking, as described in Section 3.3, can be used to limit the information about entity mentions available to our model and it is valuable in multiple ways. It can be used to simulate different scenarios, such as the presence of unseen entities, to prevent overfitting to specific entity mentions, and to focus more on context. Table 8 shows F1 scores on the TACRED validation dataset for different entity masking strategies. As we saw previously, masking with entity and grammatical role information yields the best overall performance, yielding a F1 score of 68.0. We find that using different masking strategies mostly impacts the recall, while precision tends to remain high, with the exception of the UNK masking strategy.

When applying the UNK masking strategy, which does not provide any information about the entity mention, the F1 score drops to 51.0. Using grammatical role information considerably increases performance to an F1 score of 56.1. This suggests that either the semantic role type is a very helpful feature, or its importance lies in the fact that it provides robust information on where each argument entity is positioned in the input sentence. When using NE masking, we observe a significant increase in recall, which intuitively suggests a better generalization ability of the model. Combining NE masking with grammatical role information yields only a minor gain in recall, which increases from 65.3% to 67.2%, while precision stays at 68.8%.

| Entity Masking | Precision | Recall | F1 |
|---|---|---|---|
| None | **69.5** | 58.1 | 63.3 |
| UNK | 56.9 | 46.3 | 51.0 |
| GR | 63.8 | 50.1 | 56.1 |
| NE | 68.8 | 65.3 | 67.0 |
| NE + GR | 68.8 | **67.2** | **68.0** |

Table 8: TACRED validation F1 scores for TACRED with different entity masking strategies.

### 5.4 Sample Efficiency

We expect a pre-trained language model to allow for a more sample efficient fine-tuning on the relation extraction task. To assess our model's sample efficiency, we used stratified subsampling splits of the TACRED training set with sampling ratios from 10% to 100%. We trained the previously presented model variants on each split, and evaluated them on the complete validation set using micro-averaged F1 scores, averaging the scores over 5 runs.

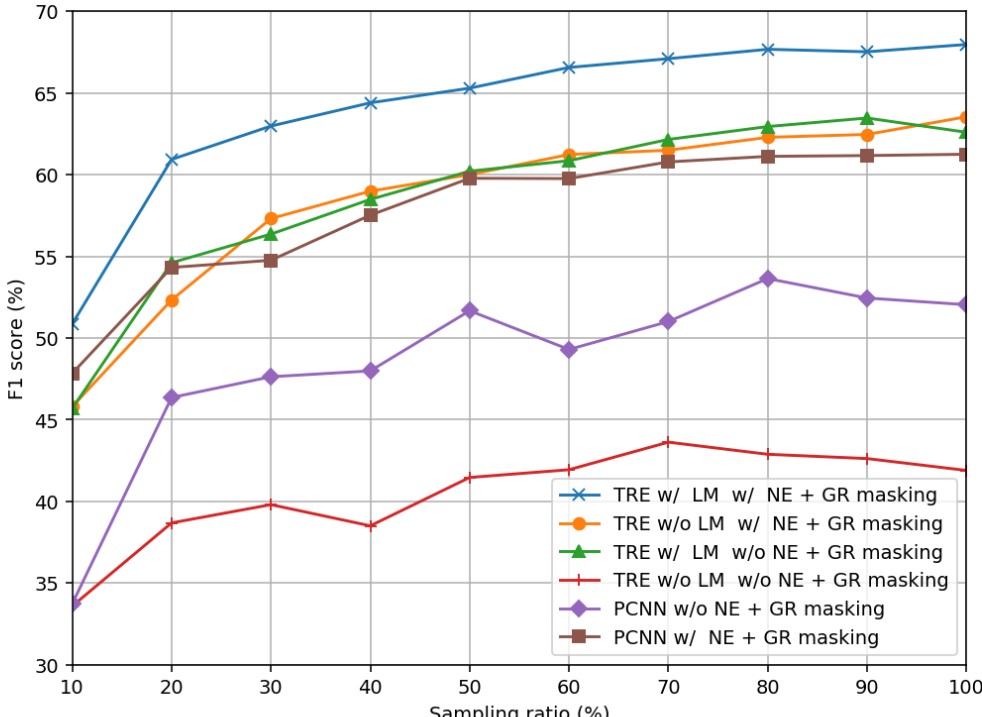

Figure 2: Micro-averaged F1 score on the validation set over increasing sampling ratios of the training set

The results are shown in Figure 2. The best performing model uses a pre-trained language model combined with NE+GR masking, and performs consistently better than the other models. There is a steep performance increase in the first part of the curve, when only a small subset of the training examples is used. The model reaches an F1 score of more than 60 with only 20% of the training data, and continues to improve with more training data.

The next best models are the TRE model without a pre-trained language model, and the TRE model without NE+GR masking. They perform very similar, which aligns well with our previous observations. The PCNN baseline performs well when masking is applied, but slightly drops in performance compared to the TRE models after 30% of training data, slowly approaching a performance plateau of around 61 F1 score. The PCNN baseline without masking performs worse, but improves steadily due to its low base score. The TRE model without a language model seems to overfit early and diminishes in performance with more than 70% training data. Interestingly, the performance of several models drops or stagnates after about 80% of the training data, which might indicate that these examples do not increase the models' regularization capabilities.

## 6. Related Work

**Relation Extraction**    Providing explicit linguistic features to inform the relation classification is a common approach in relation extraction. Initial work used statistical classifiers or kernel based methods in combination with discrete syntactic features [Zelenko et al., 2003, Mintz et al., 2009, Hendrickx et al., 2010], such as part-of-speech and named entities tags, morphological features, and WordNet hypernyms. More recently, these methods have been superseded by neural networks, applied on a sequence of input tokens to classify the relation between two entities. Sequence based methods include recurrent [Socher et al., 2012, Zhang and Wang, 2015] and convolutional [Zeng et al., 2014, 2015] neural networks. With neural models, discrete features have been superseded by distributed representations of words and syntactic features [Turian et al., 2010, Pennington et al., 2014]. Xu et al. [2015a,b] integrated shortest dependency path (SDP) information into a LSTM-based relation classification model. Considering the SDP is useful for relation classification, because it focuses on the action and agents in a sentence [Bunescu and Mooney, 2005, Socher et al., 2014]. Zhang et al. [2018] established a new state-of-the-art for relation extraction on the TACRED dataset by applying a combination of pruning and graph convolutions to the dependency tree. Recently, Verga et al. [2018] extended the Transformer architecture [Vaswani et al., 2017] by a custom architecture for biomedical named entity and relation extraction. In comparison, our approach uses pre-trained language representations and no architectural modifications between pre-training and task fine-tuning.

**Language Representations and Transfer Learning**    Deep language representations have shown to be a very effective form of unsupervised pre-training. [Peters et al., 2018] introduced embeddings from language models (ELMo), an approach to learn contextualized word representations by training a bidirectional LSTM to optimize a language model objective. Peters et al. [2018] results show that replacing static pre-trained word vectors [Mikolov et al., 2013, Pennington et al., 2014] with contextualized word representations significantly improves performance on various natural language processing tasks, such as semantic similarity, coreference resolution, and semantic role labeling. Howard and Ruder [2018] showed language representations learned by unsupervised language modeling to significantly improve text classification performance, to prevent overfitting, and to also increase sample efficiency. [Radford et al., 2018] demonstrated that general-domain pre-training and task-specific fine-tuning, which our model is based on, achieves state-of-the-art results on several question answering, text classification, textual entailment, and semantic similarity tasks.

## 7. Conclusion

We proposed TRE, a Transformer based relation extraction method that replaces explicit linguistic features, required by previous methods, with implicit features captured in pre-trained language representations. We showed that our model outperformes the state-of-the-art on two popular relation extraction datasets, TACRED and SemEval 2010 Task 8. We also found that pre-trained language representations drastically improve the sample efficiency of our approach. In our experiments we observed language representations to capture features very informative to the relation extraction task.

While our results are strong, important future work is to further investigate the linguistic features that are captured by TRE. One question of interest is the extent of syntactic structure that is captured in language representations, compared to information provided by dependency parsing. Furthermore, our generic architecture enables us to integrate additional contextual information and background knowledge about entities, which could be used to further improve performance.

## Acknowledgments

This research was partially supported by the German Federal Ministry of Education and Research through the projects DEEPLEE (01IW17001) and BBDC2 (01IS18025E), and by the German Federal Ministry of Transport and Digital Infrastructure through the project DAYSTREAM (19F2031A).

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
