# OpenReview forum: "Improving Relation Extraction by Pre-trained Language Representations"
_AKBC.ws/2019/Conference — AKBC 2019_

### Official Review · AnonReviewer3 · 2019-01-07
**Application of existing method for relation extraction**

**Rating:** 5
**Confidence:** 4

**Review:**

This article describes a novel application of Transformer networks for relation extraction.

CONS:
- Method is heavily supervised. It requires plain text sentences as input, but with clearly marked relation arguments. This information might not always be available, and might be too costly to produce manually.
Does this mean that special care has to be taken for sentences in the passive and active voice, as the position of the arguments will be interchanged?

- The method assumes the existence of a labelled dataset. However, this may not always be available.

- There are several other methods, which produce state of the art results on relation extraction, which are minimally-supervised. These methods, in my opinion, alleviate the need for huge volumes of annotated data. The added-value of the proposed method vs. minimally-supervised methods is not clear.

PROS:
- Extensive evaluation
- Article well-written
- Contributions clearly articulated

---

> ### Author Response · Authors · 2019-01-16
> **Response to Reviewer 3**
>
> We would like to thank Reviewer 3 for their review and constructive suggestions.
>
> We acknowledge that distantly- and semi-supervised methods are a vital part of (large-scale) relation extraction. In this work we explicitly focus on the supervised scenario for the following reasons:
> Our main goal is to show that pre-trained language representations, in combination with a self-attentive architecture, are able to perform comparable or better than methods relying on explicit syntactic and semantic features, which are common in current state-of-the-art relation extraction methods. Due to the automated annotation process, distantly- and semi-supervised methods introduce a considerable amount of noise. This requires extending the approach to explicitly account for the noisiness of the data, which makes it more difficult to assess the efficacy of our approach in isolation. In ongoing work we address the distantly-supervised scenario, extending our approach to account for the noise introduced during the automated annotation process.
>
> No special care has to be taken for passive and active voice. Our approach implicitly assumes the head entity to be provided first, followed by the tail entity. I.e. if the entities are provided as <Person A> <Organisation A> (assuming that person A is an employee of Organisation A), the system predicts "per:employee_of", whereas <Organisation A> <Person A> results in the inverse relation "org:top_members/employees".

---

### Official Review · AnonReviewer1 · 2019-01-09
**Incremental but solid contribution**

**Rating:** 7
**Confidence:** 3

**Review:**

This paper presents a transformer-based relation extraction model that leverages pre-training on unlabeled text with a language modeling objective.

The proposed approach is essentially an application of the OpenAI GPT to relation extraction. Although this work is rather incremental, the experiments and analysis are thorough, making it a solid contribution.

Given that the authors have already set up the entire TRE framework, it should be rather easy to adapt the same approach to BERT, and potentially raise the state of the art even further.

In terms of writing, I think the authors should reframe the paper as a direct adaptation of OpenAI GPT. In its current form, the paper implies much more novelty than it actually has, especially in the abstract and intro; I think the whole story about latent embeddings replacing manually-engineered features is quite obvious in 2019. I think the adaptation story will make the paper shorter and significantly clearer.

---

> ### Author Response · Authors · 2019-01-16
> **Response to Reviewer 1**
>
> We would like to thank Reviewer 1 for their review and constructive suggestions.
>
> We agree that adapting our approach to BERT is rather easy, in fact, we have already done so for our ongoing experiments.
>
> Regarding the last comment: We agree that our paper is a slightly adapted application of the OpenAI GPT for relation extraction. However, in the introduction, our goal was to motivate this application, because most state-of-the-art approaches (e.g. all competing approaches listed in table 4 and 5) still rely on dependency parse information and other manually-engineered features.
>
> We will rephrase our paper to clearly indicate the adaptation of the OpenAI GPT.

---

### Official Review · AnonReviewer2 · 2019-01-10
**Review of Improving Relation Extraction by Pre-trained Language Representations**

**Rating:** 6
**Confidence:** 4

**Review:**

The paper presents TRE, a Transformer based architecture for relation extraction, evaluating on two datasets - TACRED, and a commonly used Semeval dataset.

Overall the paper seems to have made reasonable choices and figured out some important details on how to get this to work in practice.  While this is a fairly straightforward idea and the paper doesn't make a huge number of innovations on the methodological side, however (it is mostly just adapting existing methods to the task of relation extraction).

One point that I think is really important to address: the paper really needs to add numbers from the Position-Aware Attention model of Zhang et. al. (e.g. the model used in the original TACRED paper).  It appears that the performance of the proposed model is not significantly better than that model.  I think that is probably fine, since this is a new-ish approach for relation extraction, getting results that are on-par with the state-of-the-art may be sufficient as a first step, but the paper really needs to be more clear about where it stands with respect to the SOTA.

---

> ### Author Response · Authors · 2019-01-16
> **Response to Reviewer 2**
>
> We would like to thank Reviewer 2 for their review and constructive suggestions.
>
> We report the best single-model performance (PA-LSTM) from the original TACRED paper in Table 4. Zhang et al. report slightly different results, 65.4 (2017) vs. 65.1 (2018), and we report the latter. The overall best performing model reported in the original TACRED paper is an ensemble combining 5 independently trained models.
>
> We will submit a revised version clearly indicating that we compare single-model performance.

---

### Public Comment · ~Roy_Fine1 · 2022-09-15
**Thanks**

Thanks. I am a sports student and I have to prepare a lot of assignments. But I always wondered which Essay writing service is the best. then I found this https://www.topessaywriting.org/samples/basketball post. This post made my work even easier and I started completing my sports assignments before their submission time. On this site, you get to see more than 100k free essay writing samples. Which makes it easy to complete your university task.

---

### Meta-Review · Area_Chair1 · 2019-02-07
**A strong model on TACRED**

**Recommendation:** Accept (Poster)
**Confidence:** 4

**Metareview:**

Current SOTA on TACRED uses precomputed syntactic and semantic features. This paper proposes to replace this pipeline with a pretrained Transformer with self-attention. This pretrained model is further fine-tuned to do the TACRED relation extraction. The reviewers like the paper and I am happy with the overall discussion. I believe the pretrained model could be useful for other relation extraction tasks, so I am accepting this with a slight reservation.

As noted by Reviewer 3, this pretrained model requires supervised annotations. It would be useful if the paper could add a discussion on the following questions:

1. Why is the supervised data required for pretraining a viable option than syntactic and semantic features? The latter are task-agnostic, so I believe they will be readily available for many languages.

2. How hard is it to create pretraining data vs. supervised relation extraction data?

---

### Decision · Program_Chairs · 2019-02-15
**AKBC 2019 Conference Decision**

Accept